# Egg Allergy: Diagnosis and Immunotherapy

**DOI:** 10.3390/ijms21145010

**Published:** 2020-07-16

**Authors:** Dulashi Withanage Dona, Cenk Suphioglu

**Affiliations:** NeuroAllergy Research Laboratory (NARL), School of Life and Environmental Sciences, Faculty of Science, Engineering and Built Environment, Deakin University, 75 Pigdons Road, Geelong 3216 VIC, Australia; awithana@deakin.edu.au

**Keywords:** hypersensitivity, recombinant allergens, immunotherapy, hypoallergens, egg allergy, allergy, egg allergens

## Abstract

Hypersensitivity or an allergy to chicken egg proteins is a predominant symptomatic condition affecting 1 in 20 children in Australia; however, an effective form of therapy has not yet been found. This occurs as the immune system of the allergic individual overreacts when in contact with egg allergens (egg proteins), triggering a complex immune response. The subsequent instantaneous inflammatory immune response is characterized by the excessive production of immunoglobulin E (IgE) antibody against the allergen, T-cell mediators and inflammation. Current allergen-specific approaches to egg allergy diagnosis and treatment lack consistency and therefore pose safety concerns among anaphylactic patients. Immunotherapy has thus far been found to be the most efficient way to treat and relieve symptoms, this includes oral immunotherapy (OIT) and sublingual immunotherapy (SLIT). A major limitation in immunotherapy, however, is the difficulty in preparing effective and safe extracts from natural allergen sources. Advances in molecular techniques allow for the production of safe and standardized recombinant and hypoallergenic egg variants by targeting the IgE-binding epitopes responsible for clinical allergic symptoms. Site-directed mutagenesis can be performed to create such safe hypoallergens for their potential use in future methods of immunotherapy, providing a feasible standardized therapeutic approach to target egg allergies safely.

## 1. An Overview

An allergy is a symptomatic overreaction by the immune system to harmless environmental substances, such as proteins from cow’s milk, fish, egg, and nuts, as well as some pollens, pet dander, and house dust mites. These substances are referred to as allergens, which are a type of antigen that prompts an intricate immune response upon contact with the immune system of the predisposed susceptible individuals. Pathogenesis involves the cross-linking of the mast and basophil-bound IgE, leading to an immediate release of allergic mediators, as a result activating T-cells, basophils and eosinophils [1]. An IgE-associated allergy is characterized as a Type 1 hypersensitivity reaction owing to the immediate inflammatory immune response that occurs upon contact with an allergen [2]. Type I hypersensitivities include life-threatening anaphylaxis, asthma, eczema, drug, insect and food allergies, which affect nearly 30% of the population worldwide [3,4]. As the severity and complexity of allergic disease increase globally, it is predominately children who suffer the burden with no optimal treatment. This unprecedented public health issue urges the need for a standardized holistic approach towards not only diagnosis and treatment but also education and research.

## 2. Food Allergy

Food allergies make up the majority of allergies today. Although global prevalence data is lacking, the World Allergy Organization (WAO), using data from 89 countries, estimates that over 250 million people are globally affected [5]. As food is essential to health and nutrition, it is yet uncertain why a food item may be identified as harmful by the body, forming the core of food allergy research. While incredible advances have been made in determining the mechanisms leading to an allergic reaction, what makes a protein within a food allergenic is still unknown.

Research is yet to fully understand the interaction between genetics, environmental factors and the protein's molecular structure that is responsible for allergic sensitization [6]. Genetics, however, may be the strongest link in further understanding this phenomenon [7]. It was found that a child has up to a 75% chance of developing an allergy given that both parents also suffer from allergies [8]. Genetically predisposed individuals have an inherited tendency to develop a food allergy with The HealthNut study re-defining “high risk” patients as having two or more allergic family members [9]. However, it is possible to develop allergies even if there is no family history, with research indicating only a slight increase in food allergy of infants with one allergic member in comparison to those with no family history [9]. This suggests other factors may also be involved, resulting in various theories, including the hygiene hypothesis [10]. This hypothesis proposes that the protective effect of microbial contact in the early stages of life is lost due to increased hygiene and extensive use of antibiotics within Western culture. Thereupon an allergy develops as the immune system lacks predisposition to infection where the microbial stimulus necessary for normal maturity of the immune system is lost [11]. A population-based study of infants suggested that, unlike peanut allergies, egg allergies are heavily motivated by environmental factors [9]. Epidemiological research indicates the hygiene hypothesis to be a complex interaction of many factors, and further large-scale population-based studies are needed to investigate genetic susceptibility, timing, allergy phenotypes and various environmental exposures [12].

The US Centers for Disease Control and Prevention (CDC) estimated in 2012 that 5.6% of children (infant to 18 years old) suffer from a food allergy [13]. An Australian study found this number to be as high as 10% in one-year-old infants, highlighting the burden of disease to be greater in young children [14]. Interestingly, challenge-proven food allergy prevalence in children under 5 years of age in Denmark is 3.6%, 4% in the United Kingdom, 6.8% in Norway and 1% in Thailand [15,16]. While high quality global data is lacking, particularly in developing countries, studies suggest food allergies have globally increased in the past decade [17]. Despite the prevalence varying from country to country, food allergies are predominantly triggered by the “Big Eight”, referring to milk, egg, tree nut, peanut, soy, wheat, fish and shellfish [14]. These account for 90% of all food-induced allergic reactions, leading to the Food Allergen Labelling and Consumer Protection Act (FALCPA) and mandating that food containing the above allergens are declared in plain language on the ingredients list. Increased awareness, improved diagnostics and pollution, in addition to the hygiene hypothesis, may explain the increase in food allergies throughout the Western countries [18]. Consequently, the rising number of individuals with food allergies has become problematic with the total cost of IgE-mediated allergies reaching more than $9.4 billion per year alone in Australia [3,5]. It is estimated that $8.3 billion of this total are indirect cost or the cost to society due to factors such as lower productivity and loss of income [5]. Globally, allergies are experienced by millions of people along with the anxiety it brings, leaving many seeking aid in conventional medications. These treatments are mostly a combination of antihistamines, immunosuppressant’s and decongestants that merely treat the symptoms of allergies. While pharmacotherapies provide temporary relief, it can, however, lead to a problematic reliance on drugs without treating the allergy itself [19]. A study in 2006 found that 1 in 5 food-allergic patients used complementary and alternative medicine with participants, even disclosing the use of homeopaths and acupuncturists to relieve symptoms [20].

## 3. Egg Allergy

The hypersensitivity to *Gallus gallus* (chicken) egg is a pervasive condition ordinarily affecting up to 9% of children worldwide [21,22]. It is documented to be one of the most prevalent food allergies among children [23]. Research conducted by the Beating Egg Allergy Trial (BEAT) in 2016 found egg allergy to be the leading cause of IgE-mediated food allergies in Australian children [22]. This was also confirmed by The HealthNut study; this cohort study found the prevalence of egg allergy at one year of age to be 9.5% when compared to other major food allergies, such as peanut (3.1%) and cow’s milk (1.5%) [24,25]. Spontaneous resolution and tolerance to egg allergy is common and occurs in 60–75% of children prior to their teenage years; however, the burden of the disease is severe during early childhood as symptoms include vomiting, abdominal pain, diarrhoea and urticaria [23,26,27]. While egg allergy is considered a childhood disease, the remaining fraction of allergic children continue to experience persistent egg allergies into adulthood, further increasing the risk of a potentially fatal reaction [24].

Egg allergy occurs as the body overreacts to proteins found in both egg white and egg yolk. The four major proteins within the egg white are the more causative agents of egg allergies, as research has found egg yolk proteins to be less allergenic (Figure 1) [28]. In 2019, Dang and associates confirmed that the majority of egg-allergic infants were sensitized to egg white allergens but not egg yolk [24]. Consequently, allergens within the egg white have been extensively studied, in contrast, egg yolk allergens have received very little attention, emphasizing the need for the holistic analysis of all major egg allergens [29]. The existing management approach to egg allergy is strict avoidance. This, however, is impractical due to the use of eggs in an extensive range of processed foods and pharmaceutical commodities, including vaccines (Table 1) [30]. Furthermore, avoidance of all egg products poses a nutritional disadvantage as eggs are of high dietetic significance, providing essential vitamins, proteins and fatty acids [31]. Contrastingly, research also indicates that the ability to tolerate cooked egg offers a potential predictor of transient egg allergy, with 80% of children with a raw egg allergy tolerating cooked forms of egg [32,33]. This is important to consider given the current management for egg allergies, highlighting the further need for accurate diagnosis, prognosis and differentiation between egg-allergic, egg-tolerant and egg-sensitized individuals.

### 3.1. Egg Yolk Allergy

Egg yolk allergies predominately affect adults, unlike egg white allergies that commonly affect young children [36]. Countless research has confirmed that a large portion of egg-allergic infants and children are sensitized to egg white but not egg yolk allergens [24]. The egg yolk contains two allergens that are much less prevalent and potent in comparison to egg white allergens. The first to be identified was a water-soluble globular glycoprotein, Chicken Serum Albumin (α-livetin or Gal d 5), followed by Yolk glycoprotein 42 (YGP42 or Gal d 6) [27]. In a cohort study, irrespective of allergy status (persistent, tolerant or sensitized), it was found that less than 8% of all infants were sensitized to Gal d 5; however, it was also observed that Gal d 5-specific IgE (sIgE) was strongly correlated with persistent egg allergy or the sensitization to multiple egg allergens [24]. As a result, clinical understanding of egg yolk allergies is limited; this in combination with the lack of standardized diagnostic reagents usher the possibility that egg yolk allergens are not currently readily recognized. Despite reports, until recently the actuality of egg yolk allergies was highly mistrusted, as it was believed that an egg yolk allergy cannot occur without an allergy to egg whites. We now know that an egg yolk allergy can sometimes be initiated by the bird-egg syndrome. As such, research is still being conducted on other egg yolk proteins that exhibit possible allergenicity, such as apovitellenin I, apovitellenin VI and phosvitin [2].

### 3.2. The Bird-Egg Syndrome

Bird-egg syndrome is a respiratory Type 1 hypersensitivity disorder [37]. It is understood that egg yolk allergy is triggered by the sensitization of the immune system to airborne egg allergens [2]. In recent years, Gal d 5 was proven to be the cross-reacting allergen responsible for the association between bird antigens and egg yolk allergies [38]. In bird-egg syndrome, the allergy to egg yolk develops succeeding the sensitization to inhalant avian antigens derived from dander, droppings, feathers and bird blood serum (Figure 2) [39]. This phenomenon is mainly seen in adults and develops as a consequence of allergic sensitization to inhalant allergens, with most patients having had regular exposure to poultry or pet birds [2]. This suggests that egg intolerance in adults is largely due to the sensitization of livetin in the egg yolk triggered by the inhalation of bird dander [39].

De Maat-Bleeker and associates in 1985 first recorded the connection between the hypersensitivity to ingested egg yolk, rhinitis and asthma in an older woman who had been exposed to a parrot [27]. Further research has found that patients suffer from both gastrointestinal and allergic symptoms, such as asthma and oedema, but—and unlike allergy to egg whites—bird-egg syndrome mostly affects adults [2,39]. In 1988, Mandallaz along with colleagues, using a radioallergosorbent test (RAST) inhibition experiment, demonstrated the inhibition of IgE-binding to livetins of hen’s egg yolk by avian antigens from different bird species [39]. This result indicates that serum proteins from different bird species contain highly conserved epitopes, which allow specific IgE antibodies to cross-react. Coexistent sensitization to other food allergens occurs in approximately 47% of egg-allergic infants; this is believed to be the reason why allergies to eggs specifically are not well documented in adults with only one case report documenting bird-egg syndrome [2,24].

This under-recognized phenomenon can be linked to occupational food allergies. Recent findings indicate occupation to be an important component in the presentation of allergic disease, mainly the development of food allergies in adults [40]. Countless occupations involve repeated respiratory and transdermal exposure to food-related allergens, resulting in sensitization. Chefs, food processing workers and even healthcare workers carry an increased risk of developing sensitization to food allergens and latex.

The importance of respiratory sensitization in adult food allergies is not well understood [39]. While symptoms of IgE-mediated food allergies such as atopic dermatitis (AD) or eczema subside through life, almost 3% of adults are still affected in industrial countries, as stress is a triggering factor [41]. A causal relationship has been suggested between AD and food allergens, yet the association between them are not yet fully understood [41]. Due to the nature of food allergies, especially in children, it is difficult to fully understand its development. Egg allergies can occur without the child ever directly ingesting the allergen; the strong association between early-onset and severe reactions, as well as the lack of sIgE in cord blood, suggests food sensitization likely occurs through an inflamed skin barrier in eczematous skin [42]. There is also evidence that AD is present prior to food allergies, establishing a link between early childhood AD and adult food allergies [41]. Airborne egg proteins, such as those available in dust samples, may predispose adults who present with the bird-egg syndrome, similarly through disruptions in the skin barrier [43]. One case of protein contact dermatitis was reported in a patient with the bird-egg syndrome in 2017 by Berbegal and colleagues [44]. Mutations in the gene filaggrin (FLG) is a major factor in the predisposition of AD; studies have found that carrying this mutation interferes with the skin barrier by 3 months of age and before the emergence of AD [45]. This further increases the risk of food allergies and supports the evidence that a disrupted skin barrier prompts sensitization [46]. This was tested in animal models with FLG-deficient mice where the application of Gal d 2 on intact skin led to the induction of cutaneous inflammation and an increase in Gal d 2 sIgE [47]. Furthermore, Flohr and colleagues demonstrated that skin impairment is not limited to visible skin inflammation but includes environmental factors, such as the use of soaps, frequency of washing and water hardness [48]. Large-scale studies are needed to further understand bird-egg syndrome and the implication of AD in relation to environmental allergen exposure to fully validate this.

### 3.3. Egg White Allergy

The egg white contains several allergenic proteins, the four major allergens being ovomucoid (Gal d 1), ovalbumin (Gal d 2), ovotransferrin (Gal d 3) and lysozyme (Gal d 4). Gal d 2 is the most abundant protein and constitutes 54% of the total egg white protein; however, Gal d 1 is found to be the most allergenic and dominant among patients (Table 2) [28]. The production of sIgE towards heat and acid-stable proteins, such as Gal d 1, is generally associated with a greater risk of systemic and or severe reactions [49]. Contrastingly, patterns in heat-labile proteins, such as Gal d 2, are commonly associated with a lower risk of severe allergic presentations. A population-based cohort study, conducted by the Murdoch Children's Research Institute, found that the ability to tolerate cooked egg offered a potential predictor in transient egg allergy, with infants who were unable to tolerate cooked eggs being five times less likely to develop tolerance [32]. The heat-resistant properties of Gal 1 have also recently been found to increase persistent egg allergy 2.5-fold, while the production of sIgE to Gal d 1-3 and 5 increased the risk of having a persistent allergy to raw eggs 4-fold [24]. While the sensitization to only one egg allergen led to a 93% resolution by age 4, these findings provide crucial clinical information as it allows for the identification of patients likely to experience a resolution to egg allergies [24]. However, the role of Gal d 3 and egg yolk allergen Gal d 5 in egg allergy sensitization has not yet been well characterized. Research has found several other minor allergenic egg white proteins that remain to be fully characterized [50].

## 4. Allergic Response

An allergic reaction or response encompasses two distinct stages based on the contact of the allergen with the immune system, referred to as the humoral or primary response and the cell-mediated or secondary response [51]. Initial contact with an allergen stimulates the humoral response as the allergen is engulfed and processed by the antigen-presenting cells (APC), presenting them to CD4^+^ naïve T-helper cells (Th0). The activation of APC allows T-cell maturation through thymic stromal lymphopoietin (TSLP). As a result of the co-stimulatory molecules expressed by the APC, Th0 cells become primed and differentiate into T-helper type 2 cells (Th2) in the presence of cytokines, such as interleukin-4 (IL-4), interleukin-25 (IL-25), interleukin-10 (IL-10) and interleukin-18 (IL-18) [52]. Th2 cells further promote the secretion of IL-4, IL-5 and interleukin-13 (IL-13). This, in turn, induces B-cells to undergo antibody class switching (differentiation) from immunoglobulin M (IgM) antibody to plasma cells secreting IgE-specific to the exposed allergen (Figure 3) [53]. The release of IL-5 by Th2 stimulates and activates the production of eosinophils. Consequently, allergen-specific IgE antibodies bind to the high-affinity FcεRI receptors on mast cells and basophils, which are the effector cells of allergies, sensitizing the immune system to the allergen [54]. The effects of these molecules are referred to as the “early phase reaction” and occurs within minutes of secondary exposure [55]. The primary response to food antigens also involves antibodies of the isotypes immunoglobulin A (IgA) and immunoglobulin G (IgG), with each differing in function and response depending on the food allergen [56]. IgG is the most common antibody, which is found in blood and other bodily fluids and protects against bacterial and viral infections. However, the involvement of IgG as an effector mechanism of food allergen-induced allergic reaction is still under question. In 2020, the WAO stated that “more studies are needed to define specific IgG as a marker of food allergy” [49].

The cell-mediated response is initiated with the secondary and subsequent contact of the sensitized immune system to the same allergen. The IgE mediated allergic response is initiated by binding of the IgE to FcεRI receptors on the surface of basophils and mast cells. The cross-linking of the FcεRI receptor occurs as one IgE molecule interacts with multiple FcεRI receptors. This results in the degranulation of mast cells and basophils, releasing pro-inflammatory mediators, while plasma cells continue to proliferate and produce excessive amounts of IgE [30]. Mediators, such as histamine, chemokines and cytokines, are released as a result of degranulation. These cell mediators stimulate various clinical symptoms, such as sneezing, itching, bronchoconstriction, rashes and even anaphylaxis (Figure 4) [57]. During the development of Type I hypersensitivity reactions, Th2 also produces novel cytokines and pro-inflammatory molecules, such as interleukin-15 (IL-15) and interleukin-31 (IL-31), in addition to traditionally discussed IL-4, IL-5, IL-9, IL-25, IL-18 and IL-13 [58,59,60,61]. A “late phase reaction” can occur up to 10 hours following the secondary exposure. This results in pro-inflammatory molecules (the cytokines, including interleukin-33 (IL-33), and leukotrienes produced in the early phase) further recruiting Th2 cells, basophils and eosinophils to the site of allergen location [55,62].

It is difficult to determine with certainty the exact window of allergic sensitization, but population-based birth cohort studies indicate that allergic sensitization occurs early-on in childhood [63]. Consequently, the process in which allergic sensitization occurs, including antigen presentation, T-cell and the immunoglobulin class switch toward IgE, is difficult to study in great detail [49]. Studies performed on animal models indicate the possibility that other cells, such as follicular dendritic cells (FDCs), are also involved in early sensitization [64]. While human data is lacking, FDCs are considered to be capable of long-term binding and present antigen-immune complexes, subsequently playing a large role in maintaining IgE memory [65]. In addition to FDCs, T follicular helper (Tfh) cells induced by migratory TSLP-activated dendritic cell (DCs) are specialized to allow B-cells to generate a stable antibody response. This occurs through the secretion of interleukin-21 (IL-21) and IL-15, inducing the differentiation of memory and plasma cells [65,66,67]. Recent progress into understanding the Tfh lineage in lymphoid organs has permitted the ongoing monitoring of this response to be more achievable in humans.

Symptoms of this response can range from mild to severe, with severity varying from one individual to the next and each reaction is unpredictable, regardless of the previous reaction [68]. Mild reactions result in minor symptoms frequently irritating the skin, causing eczema and hives. Severe symptoms, however, primarily involve the respiratory and circulatory systems and lead to difficulty in breathing, facial swelling and heart arrhythmias [69]. The severity of certain allergic symptoms can lead to anaphylaxis or even death [53].

## 5. Diagnosis of Allergy

Diagnosis begins with an extensive look at the environmental factors and symptoms experienced by the patient, followed by a physical examination to confirm an allergic reaction [70]. Often, diagnosis and treatment are made difficult, as allergies are a multifactorial condition where the symptoms and onset can often mislead medical practitioners. This is mostly due to food-induced allergenic symptoms mirroring food intolerances (non-allergic food hypersensitivity), sensitivities and other medical disorders [71]. Biological irregularity within the gastric tract results in food intolerances, like that of lactose intolerance, which is caused by the deficiency of the enzyme lactase and not mediated by IgE (a non-IgE immune response) [10]. In general, a true food allergy reaction is caused by the ingestion of a specific food with allergic symptoms arising within minutes or hours from the time of ingestion [72]. Therefore, food allergies are classified as either an IgE-mediated food allergy (IFA) or non-IgE-mediated food allergy (NFA) [73]. Initial examinations often involve elimination diets where one or more food groups are removed from the diet; this is problematic as children can often experience weight loss, eating disorders and stunted growth, leading to malnutrition [19]. Allergy testing is commonly conducted through either a skin test or a blood test, allowing the allergy specialist to conclude if the symptoms are indeed the result of an allergy, as well as identify the specific allergen [74].

In vivo allergy diagnosis is a skin test that requires the insertion of a small amount of concentrated (known) allergen into the epidermis through a needle, allowing the allergen solution to penetrate the skin [75]. Redness or blistering will occur as a result of mediators indicating that the individual tested is allergic to the introduced allergen. Severity is then assessed by calculating the diameter of the affected area and is referred to as a qualitative scoring, as there is yet no uniform way of recording this data [30]. This method of diagnosis is, however, very dangerous due to the unpredictably of its results, given that the patient is directly exposed to the adverse and sometimes severe reactions triggered by these allergens. Although very rare, fatalities have been reported as a result of skin-prick testing (SPT); for this reason, equipment and supplies are needed for subsequently treating anaphylaxis, which includes oxygen and adrenaline [30]. Another issue faced during conventional SPT is the poor representation of allergen extracts due to the biological variability of the allergen sources. As a result, these regularly fail to identify the cause of the patient's sensitivity [76]. Although recombinant allergens were first used over 20 years ago during SPTs, standardized recombinant allergen-based in vivo tests are not yet available [70]. This is despite studies verifying them to be safe, effective, specific and sensitive during allergy diagnosis [77,78].

In vitro, allergen-specific IgE testing utilizes blood tests to collect patient serum, which is then mixed with an allergen. An allergy is confirmed as sIgE binds to the allergen, becoming insoluble. A secondary anti-IgE antibody and the allergy-specific antibody concentration in the individual’s serum are used to quantify the IgE-bound fraction of the total immunoglobulin concentration [3]. This is measured through a fluorescent enzyme immunoassay (FEIA) as this laboratory technique allows for high-speed, high affinity and highly sensitive assays while using a standardized concentration of IgE in kU/I. While IgE testing can be used to diagnose a variety of allergies, it is however generally less sensitive than skin-prick testing [79]. This method of diagnosis is most useful when there is a concern regarding anaphylaxis as there is no associated risk of anaphylaxis with the blood test. It can be performed on patients taking antihistamines and other drugs, unlike the skin-prick method. However, blood-specific IgE test results can be difficult to interpret in patients with high levels of total IgE (>1000KU/L), such as in patients with eczema, given that they may be experiencing low-grade reactions simultaneously to many allergens [80].

Double-blind, placebo-controlled food challenges (DBPCFC) are the gold standard when confirming a food allergy diagnosis, although they are time-consuming and not without risk [35]. Oral food challenges (OFC) are recommended by allergists who are unable to arrive at a definite diagnosis, even after blood and skin-prick testing. DBPCFC is one of three types of oral food challenges and is widely considered to be the best as the patient receives an increasing dose of the suspected food allergen as well as a placebo (a harmless substance) [10]. The food allergen and the placebo are given to the patient separately, often in tablet form and either hours or days apart. The term “double-blind” is given as neither the patient nor the doctor will be aware as to which is given when. Food challenges such as these should only be conducted in clinical settings and administered by a medical professional who is a blind observer. Symptoms will be scored and recorded to ensure the reaction is reproducible and objective [81]. Diagnosis of an allergy still lacks clear consensus regarding the accuracy and safety of the different diagnostic methods. Consequently, the European Academy of Allergy and Clinical Immunology (EAACI) is currently in the process of developing much-needed guidelines [30].

### Molecular-Based Allergy

Molecular-based allergy (MA) or component-resolved diagnosis (CRD) is a diagnostic approach used to map and record the allergen sensitization in patients at a molecular level using allergen components (purified natural or recombinant allergenic molecules) [76,82]. The detection of the IgE antibody during the 1960s provided a precise biomarker that can be used to recognize allergic diseases triggered by environmental allergens [4]. The approach is used to detect the type of allergic reaction (i.e., IgE mediated) by determining the concentration of sIgE and total serum IgE (tIgE) [83]. This along with the utilization of DNA technology has piloted a new chapter of diagnostics, allowing allergic molecules to be characterized and cloned to determine various allergic diseases. The growing availability of allergenic molecules has introduced a new phase in diagnostics, now termed precision allergy molecular diagnostic applications (PAMD@), allowing for greater management of the allergic disease [49].

Today, many common allergenic variants have been characterized, mapped, cloned and purified. As a result, a systematic allergen nomenclature has been established to accommodate the growing number of allergens identified. The World Health Organization and International Union of Immunological Societies (WHO/IUIS) have also created an extensive database of known allergenic proteins; this list can be accessed at http://www.allergen.org. Allergens are named as per their genus and species, followed by a number to distinguish between allergens from the same species; these numbers also refer to the order of identification [76]. Allergen structure and biological function are also used to classify them into protein families, as many different molecules share common epitopes (antibody binding sites) [84]. This phenomenon is termed cross-reactivity and occurs as an IgE antibody recognizes and binds to induce an immune response to allergenic variants with similar structures from different allergen sources. Nonetheless, some proteins contain unique markers for specific allergen sources, allowing for the identification of primary sensitizers [76]. Access to natural purified or recombinant allergens has progressed our knowledge of the mechanisms leading to this phenomenon, which can vary due to the various structural, biological and physico-chemical characteristics of the allergen [63].

Molecular analysis and diagnostics are progressively used in routine care, predominantly in food allergies to improve and manage allergic patients [85]. The molecular knowledge of allergenic sensitization in patients allows medical professionals the ability to distinguish between the probability of local versus systemic reactions and the persistence of the clinical symptoms. Traditional testing and diagnostics methods are unable to establish the stability of the allergen. In contrast, CRD was used to establish that heat- and digestion-stable proteins are more likely to cause a severe allergic reaction (e.g., Gal d 1), while heat- and digestion-labile proteins (e.g., Gal d 2) more commonly cause local or milder reactions [35]. In 2019, CRD was used for the first time to examine the relevance of sIgE against egg allergens in predicting persistent and transient egg allergies [24]. The study found that exposure to egg allergens during early childhood aided in predicting the severity of egg allergy with sensitization to Gal d 1, acting as an indicator for persistent allergies. The WAO 2020 consensus document highlights the benefits of PAMD@ for patients with food allergies, food protein-induced enterocolitis syndrome, respiratory sensitization to food-pollen or inhalant food syndrome [49,86]; it allows individual patterns of IgE sensitization to be determined through the analysis of a single allergen molecules (recombinant or purified native) over complex allergenic extracts [49].

Equally, CRD can test the nature of the sensitivity specific to an allergen, thus determining if it is the result of a cross-reaction and calculating the likelihood of an allergic reaction when exposed to different allergens [76,85]. Unlike traditional SPT or in vitro-specific IgE antibody tests, molecular diagnostics can carefully individualize patients more suited to allergen-specific immunotherapy (SIT), recently also referred to as allergy immunotherapy (AIT). Consequently, the ability to measure the IgE response to certain food allergens reduce the need for food challenges. This is of great significance, especially to patients undergoing treatment, as AIT is a costly process commonly needed over long periods of time. The correct diagnosis, selection of appropriately eligible patients and the identification of the primary allergen(s) responsible for sensitization are vital for optimal cost-effective patient management [76]. Furthermore, molecular methods have been used to dismiss common misconceptions within allergy research. Food-specific immunoglobulin G4 (IgG4) was once promoted as an indicator in food-induced allergies as most patients, despite clinical confirmation, believe their symptoms were food related. Molecular diagnostics, however, recently demonstrated IgG4 to be an indicator of immunological tolerance, linked to regulatory T cells and does not induce hypersensitivity [87]. Therefore, serological testing for IgG4 is now considered irrelevant during laboratory work as it is not an indicator of food allergy but rather a physiological response by the immune system to food components. The EAACI and WAO do not recommend testing for IgG4 against food as a diagnostic tool [49,87].

The advancement of molecular techniques allows singleplex or multiplex measuring platforms to determine the production of IgE antibodies against various allergens. Singleplex uses one assay per sample and allows the clinician to select allergens based on a well-defined clinical history, as this is crucial for an accurate diagnosis. ImmunoCAP is a singleplex platform currently available, where the IgE antibodies from patient sera samples bind to immobilized allergens, and then the allergen-bound IgE antibodies are detected by fluorescence-labelled anti-IgE antibodies [88]. While the multiplex can utilize multiple assays per sample, it permits the characterization of IgE response against a comprehensive range of pre-selected allergens, independent of patient clinical history. Studies also indicate the use of multiplex assays during early life to predict the risk of developing allergic symptoms later in life [89]; this is particularly important in paediatric patients during food allergy risk assessments. PAMD@ allows the sensitization profile to be mapped using a small quantity of serum avoiding in some cases OFC, which are time consuming, costly and not free of risk. Risk assessment clinical studies to date indicate promising reliability with the greatest diagnostic accuracy for Gal d 1 (raw or heated), peanut (Ara h 6), cow’s milk (Bos 4), hazelnut (Cor a 14) and shrimp (Lit v1) [49]. Currently, the immune solid-phase allergen chip (ISAC) is the only multiplex commercially available [90]. Microarray technology allows clinicians to gain insight into the patient’s sensitization profile using a small amount of serum to identify any cross-reacting, unforeseen or potentially high-risk allergens [83]. ISAC results are analysed using a calibration curve with results recorded as ISAC Standardized Units (ISU-E) while ImmunoCAP results are provided in kilo units per litre (kU/L), meaning that these results are not comparable [88]. Thus, providing only a somewhat quantitative indication of IgE antibody levels. Additionally, ImmunoCAP measures IgE binding in conditions of excess immobilized allergens while ISAC utilizes small amounts of allergens to allow allergen-specific isotype competition and not IgE binding. Research conducted in recent years have compared results obtained through ISAC against other methods of measuring sIgE (Table 3), and it was found that assays using ISAC are reproducible but not interchangeable, the main disadvantage being a higher degree of variability as noted in low-level results of ISAC. The use of such diagnostic methods again highlights the discrepancies faced within allergy research, affirming the need for a systematic approach.

The increasing proficiency in molecular diagnostics has nonetheless provided relevant additional information, though the clinical value of many allergenic variants needs further investigation due to the rate at which new data is available [76]. 

Unmet needs within MA diagnostics include large-scale population-based studies to better define patient categories, where allergens need to be evaluated in well-characterized patients alongside healthy sensitized controls from different geographical regions. Walsh and colleagues at the Roundwell Medical Centre are the only research group to have published the cost-effectiveness on food allergy diagnostics, highlighting the need for more evaluations into the incremental benefits, comparative to the incremental cost of MA diagnostics [91]. The diagnostic accuracy of CRD-specific tests in comparison to tradition diagnosis needs to be further investigated as current studies show high specificity but low sensitivity during analysis. The systematic review conducted by Kim and associates in 2018 focus on CRD diagnosis of the “big eight” food allergies. The review summarizes the diagnostic accuracy measures emphasizing the need for a methodologically dynamic but uniform approach [92]. Coaching efforts are also needed in both clinical and research situations to advance this new era in allergology due to the immense amount of novel information currently available [76]. The current lack of relevant allergen variants needed for diagnostics assays leads to lower levels of specificity and sensitivity than that of OFC. Consequently, OFC remains the gold standard in food allergy diagnosis, limiting the progress of MA [93]. Despite some disadvantages and discrepancies, the capability to use small amounts of patient serum to isolate cross-reactive allergens and detect unknown or potentially harmful allergens are greatly benefiting allergy diagnoses, especially in patients with allergy-like symptoms (e.g. asthma, rhinitis and eczema) with the potential to revolutionize allergy diagnosis and treatment.

## 6. Treatment

Presently, there is no temporary or permanent cure for allergies. Once a diagnosis confirms an allergy, in particular an egg allergy, the current treatment involves strict dietary avoidance or minimised contact with the allergen [13]. Pharmacotherapies are used to neutralize the symptoms by blocking allergic mediators (e.g., antihistamines) but are not curative as they are unable to inhibit IgE production. Furthermore, antagonistic drugs, such as anti-histamines, anti-leukotrienes, mast cell stabiliser blockers and corticosteroids to reduce inflammation, can stimulate immunosuppression [57,75]. Life-threatening anaphylaxis is reported in 30% to 50% of all food-induced allergy cases in Australia, Asia, Europe and North America, as such epinephrine is often administered during severe reactions to prevent anaphylactic shock [6,94]. Food-induced anaphylaxis, particularly in children, occurs as a result of egg and milk allergies; while rare, fatal reactions have been reported in the United Kingdom [94].

Immunotherapy is considered to be the most efficient way to treat and relieve the symptoms of allergies, with clinical studies indicating immunotherapy approaches have the potential to not only improve allergic symptoms but to eventually prevent allergies [13,70,95]. The process involves the continued and regular administration of allergen extracts to accomplish quantifiable tolerance to the symptom-producing allergens, in patients with a discernible allergic disease [74,96]. The first principle of treatment is often referred to as desensitization, where clinical effectiveness is dose-dependent [97]. This is to say a minimal dose of an extract when administered should produce effective symptomatic control. The second being that therapeutic effectiveness increases with time as significant improvement is not commonly seen until at least three months of therapy [98]. It is not yet clear the reason behind the delayed effects of immunotherapy, however symptomatic improvement is experienced by 25% of patients regardless of the length of therapy and potency of the antigen [96]. Therefore, it is important to relay realistic expectations to patients. While systemic anaphylactic reactions are to be expected during clinical trials as well as during therapeutics, immunotherapy is considered very safe. A large-scale study conducted by Epstein in 2014 concluded that only one in 1 million injections resulted in a near-fatal or severe reaction [99].

AIT is a desensitizing therapy and the only remedial treatment available, which can be administered as an injection (subcutaneous), tablets, sprays or drops under the tongue (sublingual) [74]. This method of treatment aims to modify Th2 lymphocytes into Th1 lymphocytes and therefore inducing IgG production instead of IgE, and as such, AIT has been documented as an effective treatment for rhinitis and asthma [75]. Patients undergoing immunotherapy will need to be committed for several years for the treatment to work as well as to minimize frequent side effects. However, immunotherapy carries an inherent risk of anaphylaxis and thus extreme care and vigilance are required by the medical staff. Thus, safer immunotherapy reagents (e.g., highly purified recombinant hypoallergens) are required to minimize such anaphylaxis risks, highlighting the importance of safe recombinant allergens and hypoallergens.

The use of PAMD@ has transformed AIT as it is able to better select patients best suited to benefit from this form of therapy as well as the specific allergen(s) involved. Recent findings from clinical trials indicate that the use of PAMD@ in AIT is preferred over symptomatic forms of allergy treatment, as multiplex technologies allow for refined diagnosis and prognosis [74]. Therapy involves either sublingual or subcutaneous administration of the exact allergen(s) responsible for the clinical allergic symptoms [76]. Given well-standardized clinical and diagnostics procedures, AIT is still considered an aetiology-based treatment; its precision, however, allows for a focused and tailored approach, thus reducing costs and producing better results [49]. AIT is commonly recommended for the treatment of allergic rhinitis, sometimes even asthma due to pollen or dust mite allergy as well as for life-threatening allergic reactions to stinging insects; its use in food allergens is not yet documented [7]. To date, only major inhalant allergens containing minimal or variable amounts of minor allergens have been standardized for AIT. Therefore, AIT is not beneficial for individuals with a sole sensitization to minor allergens as successful treatment is dose dependent; thus, therapeutic success may be linked to the allergen concentration [100]. Current research indicates that venom immunotherapy can reduce the risk of severe reactions from 60% in adults to 10%, and venom therapy is presently available for bee and wasp therapy in Australia and New Zealand [7]. Subsequently, AIT is currently only recommended where symptoms are severe, the cause is difficult to avoid, such as grass pollen, and where mediations do not improve symptoms or cause adverse side effects [79]. Despite encouraging results, AIT is limited as it is time-consuming and expensive and requires strict adherence [49].

Recently, DBPCFC has confirmed high dose sublingual immunotherapy (SLIT) to be effective in reducing symptoms by over 60% [79]. Sublingual immunotherapy involves allergen extracts (drops, sprays or tablets), which are held under the tongue for a few minutes and then swallowed. A meta-analysis gathered from allergic asthma and rhinitis studies of this form of treatment confirm its safety and efficacy while reducing exacerbation rates and airway responsiveness, indicating clinical value and dose dependency [7]. Similarly, subcutaneous immunotherapy (SCIT) has been confirmed as a treatment for venom hypersensitivity, allergic asthma and allergic rhinitis. Research demonstrates a reduction in the development of new sensitizations during SCIT in patients with primary (mono-sensitized) sensitization [76]. Again, while SCIT is not readily in use for food allergies, the benefits are prolonged and persist even after years of discontinuation [101]. The disease-modifying abilities of SCIT are not without disadvantages. The use of regular injections is unpopular as it is not patient-friendly and often cause fear among children as SCIT is given over long periods of treatment and adverse reactions may occur prior to building tolerance [76]. Currently, only a few trials compare dosing regimens between SLIT and SCIT, but given that the efficacy of a broad range of allergens can be proven, SLIT is preferred by patients and parents [79]. There are also favourable reports indicating that SLIT is a promising path for the treatment of food allergies, where current treatment methods mostly rely on secondary prevention or long-term avoidance [7]. While the benefits of SLIT are clear, it still needs to be balanced against the method of delivery and the risk of anaphylaxis. However, further research is limited due to the quality and efficacy of the allergen preparations derived from natural allergen sources. For this reason, allergy diagnosis and treatment undertaking an improved approach through the use of molecular testing with well-defined, mainly recombinant allergens will permit high-resolution diagnosis [79]. 

Oral immunotherapy (OIT) refers to the ingestion of the offending allergen by an allergic individual. This approach is widely considered to be a promising treatment in food allergies with research indicating OIT is capable of modulating allergen-specific immune responses while inducing desensitization [102]. The food dosing process begins below the patient’s threshold dose; this is the minimum amount that would be required to trigger a response [103]. This dose is gradually increased over time to increase tolerance and desensitize [104]. For example, an egg-allergic patient will be given a very small amount of egg protein, and this amount will be under the threshold of what triggers an allergic reaction. The amount introduced to the patient will gradually increase over a period of months. OIT has shown to be a successful treatment for IFA but also effective in the management of NFA when using interferon-gamma (IFNy) subcutaneously [73]. Many early studies focused on the use of OIT on egg and cow’s milk allergies with initial promising results. A pilot study of egg OIT in 2007 demonstrated the desensitization of all patients at 24 months, with two participants further indicating oral tolerance during the second OFC [105]. A follow-up study based on egg sIgE adjusted the dose and duration of treatment led to all six children achieving tolerance [106]. Various other studies showing similar results of OIT have been summarized by Tang and Martino [107]. While these studies present encouraging results, they are not without bias, given the small number of participants and the exclusion of placebo (control) groups, data should be regarded with caution, especially as spontaneous resolution is common in egg-allergic patients. Progress in MA has renewed the application of OIT in food allergy treatment through use of immune-modifying adjuvants to increase tolerance. Tang and associates in 2015 conducted the first randomized placebo-controlled trial to analyse the coadministration of peanut OIT and a probiotic in children (1–10 years old). This unique approach was effectively sustaining possible “unresponsiveness” or the ability to tolerate a food after discontinuation of therapy [108]. The validity of this approach is supported by placebo-treated groups paving the way to a modified and targeted approach to OIT. 

On January 31, 2020, the U.S Food and Drug Administration (FDA) approved a standardized OIT product (Palforzia^TM^) for peanut allergy, with programs for egg and walnut allergies also announced [104]. However, OIT is not a curative therapy as the aim is to raise the threshold that can trigger an allergic reaction by providing the allergic individual protection against accidental ingestion of the allergen. While advancements are being made, OIT is not expected to allow ingestion of an allergen without limitation. A post-desensitization follow-up of a four-month controlled phase concluded that egg OIT resulted in desensitization in almost all of the participants, while tolerance was only maintained by one-third of them after a three-month period of withdrawal [109]. The validity of OIT in desensitizing patients have been confirmed in numerous randomized trials but its ability to sustain tolerance is still uncertain [107,110]. As a result, this procedure must always be conducted in a clinical setting following elimination diets and food changes to ensure safety and accuracy. Individuals who receive the therapy are also required to carry auto-injectable epinephrine and still be cautious around food.

## 7. Production and Need for Recombinant and Hypoallergenic Egg Allergens

Recombinant allergens have revolutionized allergy diagnoses since they were first made available in 1999 [111]. Numerous studies conducted have found the composition of natural allergen extracts to be severely problematic due to the presence of contaminants and undefined nonallergic materials reducing their quality [78]. Despite years of research, even today in vivo and in vitro allergy diagnoses, as well as AIT, still often use natural allergens obtained from crude sources. Complications in allergy diagnosis as a result of natural extracts have largely been overcome using recombinant allergens with well-defined purity and biological activity. Yet, currently there are no studies that document its safety and efficacy due to inconsistent quality and methods of production [78].

The production of recombinant variants of a natural allergenic protein is an option for diagnosing and treating allergies. During the process of diagnosis, it is essential to use standardized allergen extracts for consistency as well as safety to reduce the severity of individual reactions [70]. 

Recombinant proteins can also be used in treatment methods to build tolerance during AIT [58]. As individual proteins are expressed separately in bacterial and/or yeast host systems, the use of recombinant DNA technology can allow for the assembly of allergens with higher purity without contamination from other allergens. In addition, recombinant hypoallergenic variants can be produced by recombinant expression through various technologies to reduce IgE reactivity of the allergen while maintaining T-cell epitopes and immunogenicity. Therefore, inducing an allergen-specific IgG response instead of IgE response [112]. Using molecular techniques, wildtype allergens can be modified to generate allergen derivatives. By altering conformation-dependent B-cell epitopes to conserve T-cell epitopes, IgE reactivity is reduced, thus creating hypoallergens [113]. The importance in maintaining T-cell epitopes is highlighted by Smith and colleagues, who reported that children with an egg allergy have reduced function in their neonatal T-regulatory cells when compared to children without [105]. This indicates that T-regulatory cells modulate the development of food allergies and therefore T-cell epitopes must be retained for the allergen to be considered biologically active [114]. Numerous approaches to protein modification exist; these include fusion or fragmentation of molecules, random or point mutations as well at the formation of chimeras and mosaics (Table 4)—all of which can be used to reduce the risk of triggering unwanted allergic reactions.

Site-directed in vitro mutagenesis can be used to disrupt cysteine residues at multiple disulphide bonds, to produce hypoallergenic variants of a major allergen [140]. In collaboration with Drew and associates, we demonstrated the ability of mutant variants to exhibit noticeably reduced latex-allergic patient serum IgE-binding during enzyme-linked immunosorbent assay (ELISA) and immunoblot analysis [33,140]. This was further confirmed by basophil activation testing (BAT), a new sophisticated technique used for the diagnosis of an allergy to food, drugs and inhalants. It is a flow cytometry-based functional assay that assesses the degree of cell activation after exposure to stimuli. The patients’ blood is incubated with the suspected allergen and if the patient is sensitized, basophils are activated releasing chemical mediators [148]. Results obtained from BAT are very promising and appealing as it can be used to monitor immunotherapy, eliminating the need to expose patients to severe reactions [31]. Within this study, IgE reactivity of wildtype recombinant Hev b 6.01 and Hev b 6.02 was successfully demonstrated in sensitized individuals, thus capable of activating basophils. The disruption of one disulphide bridge within each allergen resulted in a decrease in serum IgE-binding and basophil activation [140]. The disruption of additional disulphide bridges led to a further decrease in serum IgE binding and basophil activation with the disruption of three or more sites, resulting in complete cancellation of basophil activation.

The same approach can be successfully applied to all major egg allergens. Conformational IgE-binding epitopes within egg allergens already directly correlated to egg allergy phenotypes are stabilized by multiple disulphide bonds (Figure 5) [149]. Mutagenesis can be used to target these cysteine residues, thereby disrupting disulphide bridging and IgE binding. As potential IgE binding is reduced, so is the risk of anaphylaxis, resulting in the development of hypoallergenic variants suitable for AIT [140]. Using this method, we successfully produced a hypoallergenic variant of the major egg allergen Gal d 1 [114]. Results within this study confirm the reduced IgE reactivity of mutant Gal d 1 in comparison to its wildtype counterpart against allergic patient serum. The reduction of IgE reactivity can be achieved either by mutation of the amino acid residues involved in IgE binding or by the disruption of the three-dimensional structure of the allergen [150]. The targeted substitution of two cysteine residues to alanine within the disulphide bonds combines both approaches. Although these preliminary results need further downstream analysis to confirm T-cell stimulation, the availability of a hypoallergenic Gal d 1 allergen would be pivotal in the treatment of persistent egg allergies.

Allergens mutated to target B-cell epitopes offer a safer alternative during immunotherapy by aiming to reduce IgE-mediated side effects [153]. While recombinant “wildtype” allergens are similar in allergenic activity to the natural allergen, they can also elicit an IgE-mediated response and thus be a major problem during AIT [154]. The approach of targeting the three-dimensional structure is based on evidence that the IgE antibody response to allergens is directed by conformational epitopes, therefore critical for IgE binding [154]. Several studies have confirmed the disruption of the three-dimensional structure results in a reduction or loss in IgE-binding capacity [155]. This can be seen in children with transient egg allergies, who produce IgE antibodies against conformational IgE-binding epitopes, which are destroyed during extensive heating or food processing [156]. Numerous hypoallergens have also been derived from the insertion of mutations into the wildtype sequence disrupting the protein structure [136]. Site-directed mutagenesis allows for the combination of both the abovementioned approaches through the mutation of cysteine residues involved in the formation of disulphide bonds. Disruption of IgE epitopes and allergen conformation can be successfully achieved by targeting the amino acids directly involved in the allergen–IgE interaction [151,152,154,157].

Unlike hypoallergens, allergoids are chemically modified or denatured native allergens which are commonly produced using various reagents, such as urea, glutaraldehyde and polyethylene glycol. These allergen derivatives are designed to retain the ability to elicit a T-cell response while minimizing the risk of the IgE-mediated response (anaphylaxis) [96]. Adjuvants are used alongside allergen extracts to boost the immunologic response in the hope of increasing efficacy. Though some studies have had success, the inability to standardize the chemical modification process limits the reliability of this approach. However, hypoallergens are considered as the best candidates for use in future specific immunotherapy regimens (Table 5). Kahler and colleagues, when measuring the basophil-activating capacity of grass pollen allergens, allergoids and hypoallergenic recombinant derivatives in 2003, found that the hypoallergenic variants showed a greater reduction in basophil activation capacity when compared to the chemically modified allergoids [158].

Research on recombinant and hypoallergenic variants are currently being continued in several directions. These include increasing the reliability of commercially available products to better standardize allergen extracts for consistent potency, shorten the dose and escalation phases, as well as the safety and efficacy of therapies available [96]. Studies are also being conducted on the use of adjuvants to elicit a more marked immune response while reducing the risk associated with immunotherapy [194]. Most recently, immunomodulatory therapies using anti-IgE and anti-cytokines are in clinical trials [96].

The major disadvantages in allergen-specific immunotherapy are associated with the potential risk of inducing anaphylaxis, the inconvenience of frequent treatments and obtaining well-defined extracts that can be tailored to an individual's patient’s sensitization profile [195]. Advancements made in allergen characterization by isolating cDNA recombinant allergens can be used to produce recombinant allergens that closely imitate the corresponding properties of the natural allergens. The following table summarizes the variations and modifications made to traditional allergen-specific immunotherapy according to target structure while highlighting the advantages of the recombinant hypoallergenic variants (Table 6).

The predominance of egg allergy and the aforementioned conditions give basis to the significance of egg allergy research and the importance of producing and using recombinant and hypoallergenic egg variants during diagnostics and treatments, as well as in the production of foods and other pharmaceutical products to better manage allergies. We aim to apply this method to all other major egg allergens.

### Recombinant Allergens in Animal Models

Itakura and colleagues in 1977 successfully expressed the first recombinant somatostatin hormone in *Escherichia coli* (*E. coli*) and, since then, *E. coli* has become the microorganism of choice for recombinant protein production [224]. The use of microbial systems to produce and express recombinant proteins has reformed biochemistry. This well-established method is now the most popular method, given the many molecular protocols and tools available, the high level of heterologous proteins produced, the immense catalogue of expression plasmids, engineered strains and methods of cultivation [225]. Given such an efficient and popular expression platform, animal models are rarely used. However, as more is discovered about T-cell epitopes, peptides of major allergens are being produced to decrease the risk of IgE-mediated reactions while retaining immunogenicity [202]. Consequently, researchers have been able to use mouse models to monitor the response of Fed d 1, the major cat allergen, and Der p 1 and Der f 1 dust mite allergens to explore the ability of peptides in reducing T-cell-dependent immune responses [202].

As previously stated, future AIT may take several forms, including mucoadhesives, allergoids and modified allergens. Frew and associates in 2017 produced promising experimental data in mouse models when allergens were delivered via the mucosa. This method allows for smaller amounts of an allergen to be administered and as a result reduces the risk of local side effects. Results also indicate that adjuvants derived from T-cells enhanced the efficacy of SLIT in a mouse model of asthma [226]. Other novel approaches to SLIT are said to have shown favourable results in mouse models when using genetically engineered hypoallergens [227]. In 2016, Ilaria used the TLR5 ligand (a fusion protein of flagellin) for intranasal and intraperitoneal inoculation of food allergy in mouse models, which decreased IgE production but did not lead to allergic sensitization [228].

While mouse models have substantiated some results, mainly in asthma, indicating the possibility of inducing T-cell tolerance, results thus far still have only proven successful when administered via the mucosa [96]. As a result, allergic patients generated divisive results, thus further supporting the use of bacterial vectors and expression systems as the preferred method of recombinant protein expression.

## 8. Current and Future Direction of Egg Allergy Research

The unmet medical need for an effective form of therapy for food allergy is extremely problematic for the Western world. The advances in the identification of allergens at a molecular level have revolutionized allergy diagnostics; PAMD@ has allowed for establishing specific allergens associated with diverse risk profiles [49]. This has allowed for several therapeutic approaches for both food allergen-specific and non-specific allergens, limiting the exposure to OFC and therefore advert anaphylactic reactions affiliated with diagnosis. Currently, the exploration for the most prevalent food allergens, its diagnosis and treatment are underway, yet there are no accepted therapies proven to desensitize food allergic patients without the guarantee of unintentional exposure [229]. In addition, while methods such as PAMD@ is advantageous, providing data on cross-reactivity, primary sensitization and risk assessments, the limited availability and associated costs have restrained its use and recognition [49].

Globally allergic disease is reaching epidemic levels; however, the severity of this is often misdiagnosed, under-recognized or maltreated due to IgE-mediated symptoms being similar to that of other conditions [76]. Accordingly, allergy education and training in molecular diagnostics methods need to be widely available. The next phase in allergy diagnosis and treatment is personalized medicine, which separates patients by phenotypes and endotypes (asthma, severe asthma and rhinitis) [49]—further indicating the need for physicians to be trained in the proper identification of allergens inducing primary sensitization or cross-reactivity [49].

In 2019, CRD was successfully used to identify sIgE to Gal d 1 and Gal 2, which was indicative of a persistent egg allergy, and this information was used to accurately predict the phenotype of egg allergy with 95% specificity and 45% sensitivity [24]. Evidence suggests that patients with a transient egg allergy will have the most favourable response to therapy, while persistent food allergy may be more challenging due to the need for prolonged treatment, failure to develop tolerance and more serious adverse reactions while on therapy [24]. For this reason, as experience with these various forms of treatment regimens increase, physicians will need to be able to counsel and optimise therapeutics to the individual needs of their patients.

The current knowledge in understanding the molecular process of allergies not only allows the practitioner to more precisely tailor treatment towards each individual's specific needs but also determines the most effective one given individual circumstances. Differentiating between major and minor allergens allows for the improved selection of patients for AIT, as AIT is more effective against major allergens [49]. AIT is currently considered the forerunner for personalized precision medicine; however, AIT is better suited for inhalant allergies as patients without cross-reactivity respond better to treatment and thus the need for further improvement with food allergens, which often cross-react [230]. Allergen-specific methodologies for food allergens include oral, sublingual and epicutaneous immunotherapy, using native food allergens, mutated recombinant proteins or excessively heated food. OIT is one of the most vigorously investigated therapeutic approaches of food allergy as it has been shown to desensitise the patient to eggs, milk and peanuts. Analysis of current studies and trials indicate OIT to be the most effective form of desensitization in patients [110]. Multiple OIT using egg-allergic individuals show tolerance with a total of 87% reaching the goal egg dosage [109]. Recently, the co-administration of OIT with a bacterial adjuvant was suggested as a potential treatment for food allergy [108]. However, researchers are still unable to show the development of tolerance and, as such, the possibility of permeant tolerance is still widely debated [156]. Non-specific approaches include monoclonal anti-IgE antibodies in the hope of increasing the threshold dose for reactivity [229].

The recent developments in purified native allergens, recombinant allergens and hypoallergens in combination with PAMD@ have provided extensive knowledge in allergic disease and tolerance. While AIT has been used for over 100 years, the new data regarding peripheral T-cell tolerance via T regulatory cells are key in suppressing allergic inflammation, providing a possible curative and targeted approach to allergic disease [58]. SLIT and SCIT are the two main routes of administration for immunotherapy, but SLIT is considered to be more favourable with 50% of patients who underwent DBPCFC to standardized hazelnut extract building tolerance at 12 weeks [76]. The availability of safe and standardized allergen extracts in the future will allow more patients to self-administer at home, depending on the severity; only the first dose may need to be supervised in a clinical setting. This would greatly reduce the time-consuming nature of current treatments on both the patient and the practitioner, as well the burden of costs involved [231].

Advances in molecular diagnostics will ensure the development of safe and effective diagnostic methods and immunotherapy reagents with high pharmaceutical quality. The ultimate goal of treatment is permanent tolerance. This can only be established when the food can be ingested without allergic symptoms, and without the need for concerns over continuous exposure. In comparison, it has been observed that protection of the desensitised state is dependent on the regular ingestion of the allergen and the discontinuation or interruption of dosing may lead to loss of protection. This was indicated by the results with patients re-developing symptoms, highlighting the importance of finding ways to achieve permanent tolerance [232]. Currently, there is no data that can formally exclude spontaneous resolution, or other possibilities, that may lead to such promising results of OIT or any other therapies of egg allergies. Studies thus far are limited not only by the lack of standardized diagnosis and treatment methods, but also by the lack of safer and well-standardised IgE-reactive recombinant allergens and their hypoallergenic variants, as well as by the lack of longitudinal population-based studies with a holistic approach to egg allergies.

## 9. Conclusions

As a result of the problems associated with the inconsistent quality of the natural-based allergen extract products currently available for in vivo and in vitro diagnosis and immunotherapy, they do not meet international standards for medical products [78]. Despite the growing availability of refined investigative technologies, these problems cannot be easily overcome due to the inherent complications associated with allergen sources and methods of extraction [112].

The production of hypoallergenic variants of recombinant egg allergens allows for downstream applications in safe and effective therapeutics, not only for the Australian population but globally. Countless research teams globally hope to find answers for allergy sufferers. The development of hypoallergenic variants of egg allergens are given utmost priority by many researchers, as hypoallergens are uncontaminated by other egg proteins and therefore can be used to desensitize egg-allergic patients without the risk of adverse allergic reactions.

Recombinant and hypoallergenic variants for all of the major egg allergens will also further allow for molecular diagnostics to replace SPT. This can be used as a first-line diagnostic tool to reduce the diagnosis process, as well as to provide a complete analysis of individual IgE profiles. This can be used to predict the evolution of allergies and its risk, eliminating the need for OFC, the current gold standard in allergy diagnosis [230]. All of the research conducted on hypoallergens and IgE-reactive recombinant allergen production still requires clinical trials and safety assessments prior to patient administration or use as an in vivo diagnostic tool. As our understanding of allergenic molecules and their role in sensitization is cultivated, we can expect profound progress in allergy treatment and diagnosis, increasing the quality-of-life in egg allergy sufferers.

## Figures and Tables

**Figure 1 ijms-21-05010-f001:**
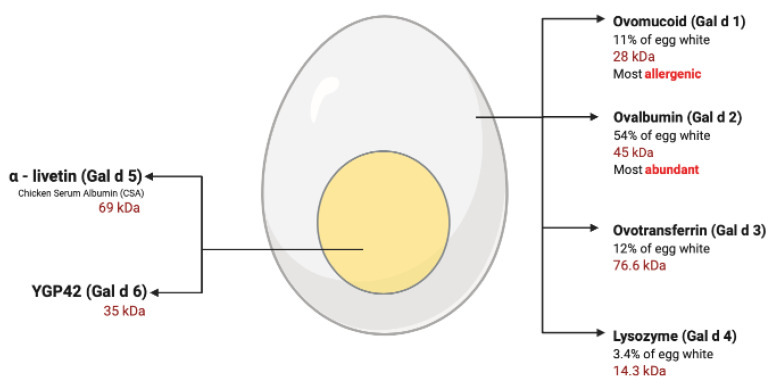
Egg allergens. This figure highlights all six major egg allergens with some of their physicochemical properties (figure adapted from Dhanapala et al. [34]).

**Figure 2 ijms-21-05010-f002:**
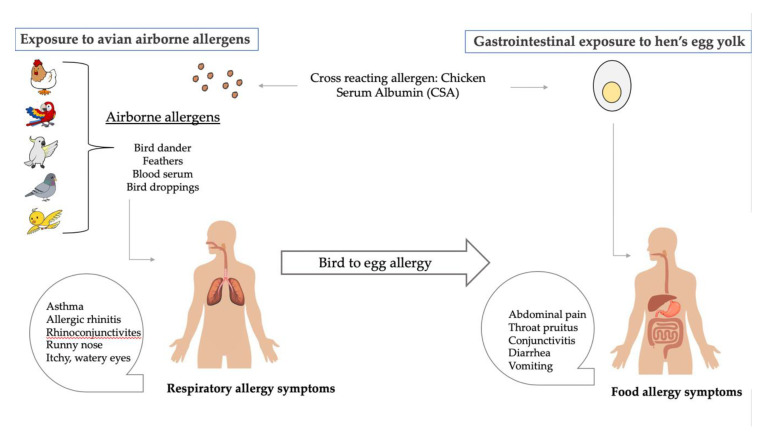
An illustration of the bird-egg syndrome. An individual develops an allergy to hen’s egg yolk following exposure to birds. Sensitisation to inhalant avian allergens occurs, resulting in respiratory allergy symptoms. The cross-reactive allergen that is responsible for producing both respiratory and gastrointestinal allergy symptoms in bird-egg syndrome is identified as Gal d 5 (α-livetin/chicken serum albumin), a type of serum albumin present in birds and the egg yolk as well (figure adapted from Dhanapala et al. [34]).

**Figure 3 ijms-21-05010-f003:**
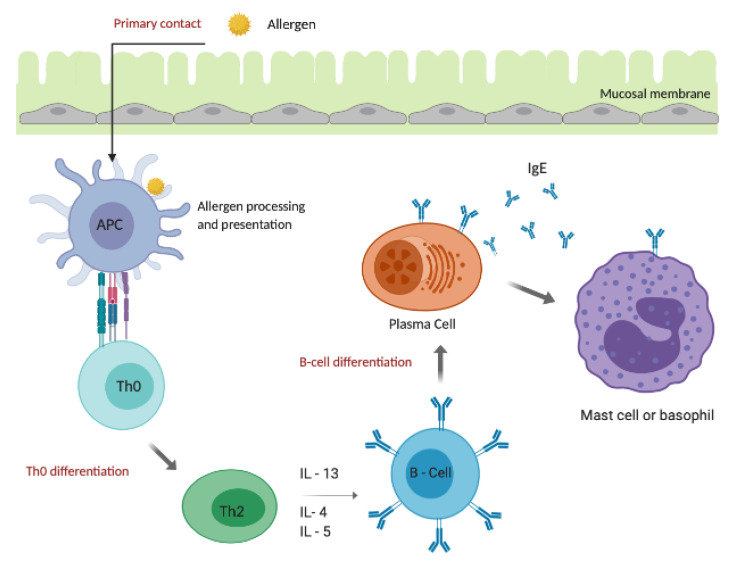
Diagram of the humoral allergic response. This figure outlines the steps of the humoral response. Once the allergen is processed and presented by the APC’s to the Th0 cells, the Th0 cells differentiate into Th2 cells in allergic individuals. Th2 secrete IL-4, IL-5 and IL-13, triggering B-cell differentiation into plasma cells. As a result, they proliferate and produce excessive amounts of allergen-specific IgE that bind to FcεRI receptors on mast cells and basophils. This process is said to sensitize the immune system. Secondary and subsequent exposure to the same allergen (see Figure 3) leads to cross-linking at the FcεRI receptors, triggering the release of mediators such as histamine, causing an allergic inflammation.

**Figure 4 ijms-21-05010-f004:**
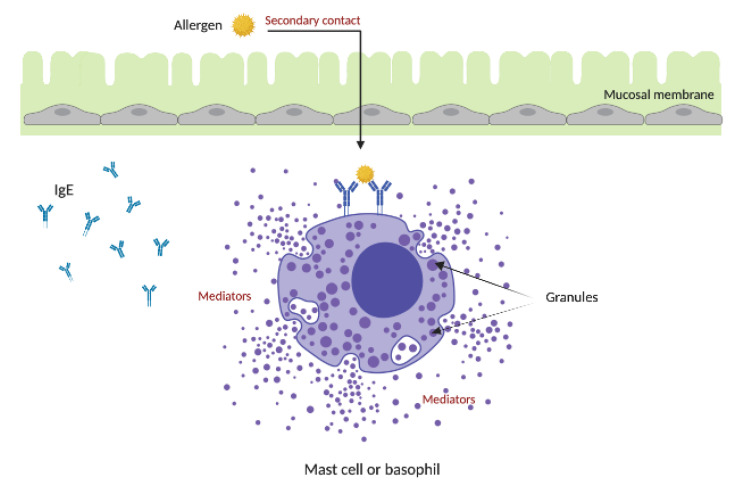
Diagram of the allergic response. Upon secondary contact by the immune system to the same allergen, the allergen binds to multiple IgE on mast cells and basophils. Cross-linking of the FcεRI receptors trigger the release of the mediators, including histamines, which are responsible for the symptoms of allergic response.

**Figure 5 ijms-21-05010-f005:**
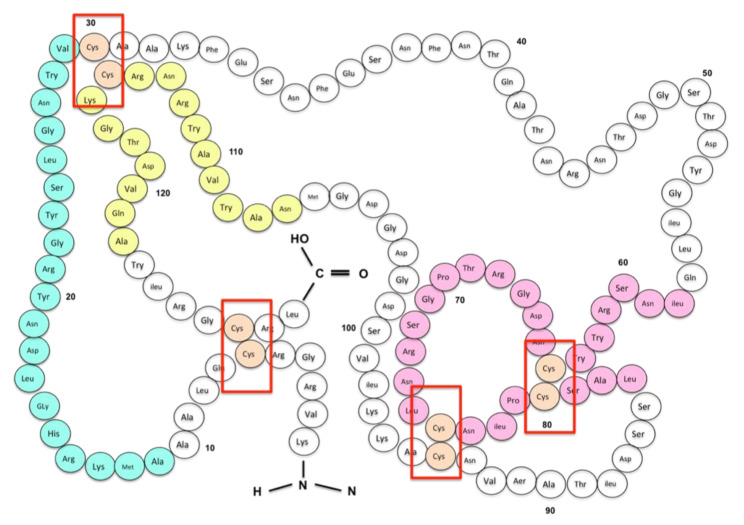
Lysozyme (Gal d 4) structure. This figure highlights in colour the three IgE-binding epitopes and cysteine residues that connect the four disulphide bridges. The red frames allow for the clear visualisation of the interaction between the IgE-binding epitopes and disulphide bridges [151,152].

**Table 1 ijms-21-05010-t001:** What to look out for in foods that may contain egg allergens (table adapted from Caubet and Wang [35]).

Avoid Foods Containing These Ingredients	Egg Proteins Can Be Found in
Albumin (can be spelled “albumen”) Egg (these can be listed as dried, powdered, solids, white and yolk) Eggnog Globulin Lysozyme Mayonnaise Meringue (meringue powder) Ovalbumin Ovovitellin Surimi	Macaroni Marzipan Marshmallows Nougat Pasta Baked goods Egg substitutes Lecithin

**Table 2 ijms-21-05010-t002:** Summary of the four main egg white allergens and their properties (table adapted from Caubet and Wang [35]). + Low; ++ Moderate; +++ High.

Allergen	Egg White (%)	Molecular Weight (SDS-PAGE)	Carbohydrate (%)	IgE-Binding Activity (Digestive)		Allergenic Activity
				Heat-Treated	Enzyme-Treated	
Gal d 1	11	28 kDa	25	Stable		+++
Gal d 2	54	45 kDa	3	Unstable		++
Gal d 3	12	76.6 kDa	2.6	Unstable		+
Gal d 4	3.4	14.3 kDa	0	Unstable		++

**Table 3 ijms-21-05010-t003:** Immune solid-phase allergen chip (ISAC), ImmunoCAP and skin-prick testing (SPT): The advantages and disadvantages (table adapted from the World Allergy Organization GA^2^ LEN consensus document [76]).

	Advantages	Disadvantages
**ISAC**	No interference due to high total IgE Less allergen needed per assay 112 allergens can be analysed in parallel Natural and recombinant proteins are compatible 30 μL of serum or plasma	Manual method Less sensitive Not recommended for monitoring sensitization May cause interference between IgE and IgG Semi-quantitative assay
**ImmunoCAP**	Appropriate for monitoring sensitization High sensitivity Quantitative analysis Natural and recombinant proteins, as well as crude extracts, can be applied	40 μL of serum needed per allergen One allergen per assay Detects low-affinity antibodies which may not be clinically relevant
**Skin Prick Test (SPT)**	Prompt readings Extract-dependent high sensitivity	One allergen per prick Only crude extracts used Manual Cannot be used for monitoring sensitization

**Table 4 ijms-21-05010-t004:** Mutational approaches used to produce recombinant hypoallergens (table adapted from Tscheppe et al. [113]).

Strategy	Definition	Allergen Source	Molecule/s
Fragmentation	The cDNA coding for a specific allergen is fragmented into ≥ 2 parts; the fragments may overlap or express individually	Birch pollen	Bet v 1 [115,116]
		Cow dander	Bos d 2 [117]
		Storage mite *(L. destructor)*	Lep d 2 [118]
		Timothy grass pollen	PhI p 1 [119]
		House dust mite (*D. pteronyssinus)*	Der p 2 [120]
Oligomerization	≥2 copies of the allergen-encoding cDNA are linked by oligonucleotide spacers with an open reading frame allowing the complete construct to express	Birch pollen	Bet v 1 [121]
Mosaics	The cDNA coding for a specific allergen is fragmented into several parts and the fragments are re-joined in an order different to the original sequence. If the sequence parts originate from >1 allergen the resulting protein is regarded as a hybrid mosaic	Birch pollen	Bet v 1 [122]
		Timothy grass pollen	PhI p 1 [123] PhI p 2 [124]
		Cat	Fel d 1 [125]
Chimeras/allergen hybrids	Chimeric/hybrid proteins are created by joining the genetic information of ≥2 different proteins such as constructs may contain parts or the complete original proteins	House dust mite (*D. pteronyssinus)*	Der p 1 [126] Der p 2 [127]
		Timothy grass pollen	PhI p 1 [128] PhI p 2 [128,129] PhI p 5 [128] PhI p 6 [129]
		Yellowjacket	Ves v 5 [130]
		Paper wasp	Pol a 5 [130]
		Honeybee	Api m 1 [131] Api m 2 [131] Api m 3 [131]
		Japanese cedar	Cry j 1 [132] Cry j 2 [132]
Point mutations	One or more nucleotide triplets coding for a specific amino acid is/are altered to replace the original amino acid at its exact position by an amino acid with different physicochemical characteristics	Birch pollen	Bet v 1 [133,134] Bet v 4 [135]
		*Brassica rapa* pollen	Bra r 1 [136]
		Carp	Cyp c 1 [137]
		*P. Judaica* pollen	Par j 1 [138]
		*Gallus gallus* (Ovomucoid)	Gal d 1 [114,139]
		Latex *H. brasiliensis*	Hev b 6.02 [140,141] Hev b 5 [142]
		House dust mite (*D. pteronyssinus)*	Der p 2 [143]
		Peanut	Ara h 1-3 [144,145,146]
		Ryegrass pollen	Lol p 5 [147]

**Table 5 ijms-21-05010-t005:** Overview of allergen derivatives developed for immunotherapy and its features (table adapted from Marth et al. [159]).

	Recombinant Wildtype Allergens [129,160,161,162,163,164]	Derivatives of Recombinant Allergens [116,165,166,167,168,169,170,171,172,173,174,175,176,177,178]	T-cell Peptides [179,180,181,182,183,184,185]	Peptide Carrier Fusion Proteins [186,187,188,189,190,191,192,193]
**Immune response:**				
IgE reactive	+	+/−	−	−
T-cell reactive	+	+	+	−
Induce protective antibodies	+	+	−	+
**Possible side effects:**				
IgE	+	−	−	−
T-cell mediated	+	+	+	−

+ Induces a response; − does not induce a response; +/− can induce a response/allergen dependent; derivatives of recombinant allergens, including mutants, fragments and oligomers.

**Table 6 ijms-21-05010-t006:** Summary of variations made to traditional allergen-specific immunotherapy (table adapted from Valenta et al. [195].

Target	Type of Modification	Advantage
Antigen	Chemically modified allergen extracts (allergoids, haptens, PEG)	Allergenic activity is reduced, inducing tolerance and Th1 responses [163,166,196,197]
	Recombinant allergens and recombinant hypoallergens	Allergen specificity; induces blocking of antibodies and increases safety Immunomodulation occurs with the induction of T-cell tolerance [160,198,199]
	T cell peptides	Induction of T-cell tolerance increases allergen specificity and increases safety [200,201,202]
	B cell peptides	Induction of blocking antibodies increases allergen specificity and increases safety [150,186]
	Mimotopes	Allergen specificity induces the blocking of antibodies in DNA vaccines [203,204,205]
	DNA Vaccines	Induction of Th1 response leads to allergen specificity [150,206,207]
Route/mode of administration	Oral/sublingual administration	Safety. The induction of T-cell anergy makes the treatment easy to perform and convenient for patients [208,209,210]
	Nasal administration	Safe and convenient for patients [211,212]
Adjuvant	Al(OH)_3_	Reduces anaphylactic side effects [213,214]
	CpG, MPL liposomes	CpG reduces allergen activity Overall induces Th1 [215,216]
	Chitosan-nanoparticles	Induction of T-cell tolerance [215,217]
	Carbohydrate based particles	Ease of production and reduced tissue damage [206,218]
	Live vaccines	Induces Th1 response [219,220,221]
	Surface layers	Induces Th1 response [222,223]

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
