# Peer review of "Egg Allergy: Diagnosis and Immunotherapy"

_ijms, 2020, doi:10.3390/ijms21145010_

Round 1

Reviewer 1 Report

This article is a comprehensive review on allergies in which the authors carefully describe the allergic response, the different diagnostic methods and treatments currently used in allergic patients. I really like the review; however, I think it should be reorganized in some way.

General comments

Point 1 is about allergies in general and in point 2 the authors focus on food allergies, especially egg allergy. However, in point 4, where they discussed Molecular-based allergy, and in point 5, Production of recombinant and hypoallergenic egg allergens, they write again about allergies in general. I proposed the authors to change the order of these parts of the review to improve the message.

On the other hand, point 6 about Immunotherapy of allergic disease should also be rewritten. The review has a part in which the treatment in allergic diseases is discussed (point 1.3) and it sounds strange to have a point about immunotherapy later (point 6). Perhaps point 6 should be divided between points 1.3 and 5, where recombinant and hypoallergenic allergens are discussed.

Finally, despite having a very attractive title, I have doubts about it. I feel it doesn´t represent the overall message of the review.

Specific comments:

  1. Page 1, line 11: “condition” should not be bold.
  2. This reviewer suggests changing the phrase “The subsequent instantaneous inflammatory….” in the abstract. The response is not only characterized by the excessive production of IgE.
  3. Page 1, line 17: Specify FOOD allergies.
  4. Page 2, line21: change the word where for
  5. Page 3, line 15: change the word material for another one, e.g. allergen.
  6. Page 4, line 1: correct guildline.
  7. Page 4, line 5: allergy-specific or allergen-specific?
  8. Page 12, Table 4: references should be included in the table.
  9. Page 13, line 15: correct the words durimg and therapeutics.
  10. Page 13, line 23: correct the word therpies.
  11. Page 14, tables 5 and 6: references should be included in the tables.

Author Response

We thank you for your careful reading of our manuscript, kind words and constructive feedback. We have taken the comments on board to improve, clarify and restructure the manuscript. Please find below a detailed point by point response to all the general comments (reviewers’ comments in blue, our response in black) and specific comments. Please note all of the specific comments were rectified, they are listed below and highlighted in yellow in the manuscript. Changes made for general comments were also ‘tracked’ and highlighted in blue. Due to major changes to the structure of the manuscript, a brief explanation is also provided below.

  • Page 1, line 72: The word “condition” is no longer bold
  • The phrase “The subsequent instantaneous inflammatory….” in the abstract has been changed to highlight that the response is not only characterized by the excessive production of IgE.
  • Page 1, line 12: This sentence in the abstract was changed entirely and now specifies egg allergens.
  • Page 7, line242:  The word wherewas removed
  • Page 11, line 380: The word material changed to allergenic variant
  • Page 10, line 342: Corrected the word guideline.
  • Page 10, line 344: Allergen-specific is the correct term
  • Table 4, 5 &6: References now included in the table.
  • Page 13, line 468: Corrected the words duringand therapeutics.
  • Page 13, line 479: Corrected the word therapy

General comments

Point 1 is about allergies in general and in point 2 the authors focus on food allergies, especially egg allergy. However, in point 4, where they discussed molecular-based allergy, and in point 5, Production of recombinant and hypoallergenic egg allergens, they write again about allergies in general. I proposed the authors to change the order of these parts of the review to improve the message.

            We have reorganised the manuscript to improve our message of egg allergy importance and need for standardised diagnosis and treatment.  Point 1 now gives a brief introduction into allergies, food allergy is now discussed as a subsection under general allergies. Point 2 now focuses on egg allergies specifically, previously this point was discussed after the ‘allergic response’.  The allergic response is now discussed in point 3, followed by ‘diagnosis of allergy’ (point 4). This point was re-structured to also discuss molecular-based allergy as a diagnostic approach as a subsection. Treatment is now point 5 and 6 is now the ‘production and need for recombinant and hypoallergenic egg allergens’. We believe these changes allow better clarification within the manuscript but still believe allergy and food allergy should be discussed holistically prior to discussing egg allergens. Again, discussing the lack of standardized protocol as a whole highlights the need for consistency during diagnosis and treatment.

On the other hand, point 6 about Immunotherapy of allergic disease should also be rewritten. The review has a part in which the treatment in allergic diseases is discussed (point 1.3) and it sounds strange to have a point about immunotherapy later (point 6). Perhaps point 6 should be divided between points 1.3 and 5, where recombinant and hypoallergenic allergens are discussed.

            The ‘immunology of allergic disease’ has now been removed from the manuscript and the information reorganized and re-written into ‘treatments’ in point 5 to better discuss the importance of immunotherapy. The current research on immunotherapy was merged with ‘production and need for recombinant and hypoallergenic egg allergens’ in point 6. We feel these changes have allowed for a better-structured manuscript with a clearer message.

Finally, despite having a very attractive title, I have doubts about it. I feel it doesn´t represent the overall message of the review.

            The new title is “Egg allergy: diagnosis and immunotherapy”. We agree that while the title was attractive this new title best represents the overall message of the review.

Reviewer 2 Report

The beginning of the submission appears to be written by an undergraduate who has yet to become accomplished with the subject of allergy, the English language or the use of spell check. The final sections are from a dated treatise on the generic use of molecular diagnosis and modified allergens with random reference to old studies on egg allergy. Neither author seems very interested in the central section on immunotherapy or the unmentioned T cell.

The Murdoch prevalence data is controversial and the follow up of the same children cast considerable doubt on the validity of test done on the 12-month-old children. Regardless of the views of the authors about the study is not just be glibly used as the final statement on prevalence without reference to other work and to real-life allergy.

Oral immunotherapy and its long-term effectiveness is still a major and topic in food allergy and this includes egg allergy. It is covered in high profile papers not mentioned here and is quite different to sublingual immunotherapy.

There are some interesting and informative studies published on component resolved diagnosis for egg allergy including one from Melbourne. A review should take the trouble to find out about them and analyses them. Any focus on egg allergens needs to describe the relative importance of the different components.

Author Response

Thank you for taking the time to review our manuscript and providing feedback. In response to your comments major changes were made to the entire manuscript. In addition to major changes, the following information was also edited or included. Due to major changes to the structure of the manuscript, please find below a brief points to all the general comments (reviewers’ comments in blue, our response in black).

The beginning of the submission appears to be written by an undergraduate who has yet to become accomplished with the subject of allergy, the English language or the use of spell check. The final sections are from a dated treatise on the generic use of molecular diagnosis and modified allergens with random reference to old studies on egg allergy. Neither author seems very interested in the central section on immunotherapy or the unmentioned T cell.

The Murdoch prevalence data is controversial and the follow up of the same children cast considerable doubt on the validity of test done on the 12-month-old children. Regardless of the views of the authors about the study is not just be glibly used as the final statement on prevalence without reference to other work and to real-life allergy.

Oral immunotherapy and its long-term effectiveness is still a major and topic in food allergy and this includes egg allergy. It is covered in high profile papers not mentioned here and is quite different to sublingual immunotherapy.

There are some interesting and informative studies published on component resolved diagnosis for egg allergy including one from Melbourne. A review should take the trouble to find out about them and analyses them. Any focus on egg allergens needs to describe the relative importance of the different components.

The changes are tracked and highlighted and underlined in red within the text of the manuscript.

  • The abstract was re-written, and the beginning of the manuscript reorganised.
  • The entire manuscript was thoroughly checked for spelling mistakes
  • The Murdoch prevalence data is now updated by 2019 HealthNut and 2016 BEAT study data
  • The immunotherapy section reorganised and rewritten
  • The final sections were rewritten using WAO 2020 data
  • The role of T-cells is discussed further
  • The molecular diagnostic sections discuss egg allergy data specifically with the use of CRD
  • Oral immunotherapy is further discussed

Reviewer 3 Report

This is an important review on egg allergy, diagnosis, and immunotherapy. Several points should be discussed as follows:

1.Why do egg yoak antigens or egg white antigens preferably cause allergy in adults or children, respectively? Are there any possible reasons for the phenomena reported?

2.Why is ovomucoid more highly antigenic compared to ovalbumin? Is it just due to its stability against heat or digestion, or any other reason? Please comment on this point.

3.In bird-egg syndrome (Fig 4), can percutaneous sensitization of airborne allergens occur as a possible mechanism of sensitization, especially in skin barrier-disrupted humans like atopic dermatitis patients?

Author Response

We thank you for taking the time to read our manuscript, we appreciate your helpful feedback and kind words. We have taken the comments and suggestions on board to improve and clarify the manuscript. We have added further detail to clarify the questions listed below. Due to major changes to the structure of the manuscript, please find below a brief response to all the general comments (reviewers’ comments in blue, our response in black). Changes made for general comments were also ‘tracked’ and highlighted in green.

  1. Why do egg yolk antigens or egg white antigens preferably cause allergy in adults or children, respectively? Are there any possible reasons for the phenomena reported?

Egg allergy is the most common allergy among young children and extensively studied; however, this often resolves in almost 75% of patients during childhood. Egg allergy in adults generally begins during childhood with severe systemic allergic symptoms. The role of Gal d 3 and Gal d 5 in sensitization is not yet well understood but Gal d 5 is linked to persistent egg allergies. Contrastingly adult-onset egg allergy has rarely been mentioned in the literature and therefore there is no exact reason for this. However, given that Gal d 5 is common in patients with persistent egg allegies, that individuals often are allergic to more than one food allergen and the lack of standardized diagnostic approached that egg yolk allergy is possible that it is often missed during diagnosis in adults. Gal d 5 is also heat liable, again potentially resulting in tolerance or very mild reaction by individuals. Bird-egg syndrome, however, is where patients are sensitized to Gal d 5 via airborne bird allergens. This has been further clarified throughout the manuscript.

  1. Why is ovomucoid more highly antigenic compared to ovalbumin? Is it just due to its stability against heat or digestion or any other reason? Please comment on this point.

Gal d 1 only makes up 11% of the egg white but due to its heat resistant properties, it is also the most allergenic compared to that of heat liable Gal d 2. Gal d 1 is used as a diagnostic marker for egg allergies, therefore it is considered to be the dominant egg allergen. This has been further clarified within the manuscript using the most recent data and all changes tracked and highlighted in green.

  1. In bird-egg syndrome (Fig 4), can percutaneous sensitization of airborne allergens occur as a possible mechanism of sensitization, especially in skin barrier-disrupted humans like atopic dermatitis patients?

Very little research has focused on the egg yolk allergens and as a result, the bird-egg syndrome is not yet fully understood. However, airborne egg proteins such as those available in dust samples may predispose adults who present with the bird-egg syndrome similarly through disruptions in the skin barrier. There is also evidence that atopic dermatitis is present prior to food allergies establishing a link between early childhood atopic dermatitis and adult food allergies. This point, as well as the FLG mutation, are now discussed in detail on the manuscript, all changes tracked and highlighted in green.

Round 2

Reviewer 1 Report

As a reviewer, this rewritten version of the manuscript has been much heavier to read than the first one. I think the authors should summarize the review a lot. I recommend focusing solely on what's new.

Author Response

We thank you for your careful second reading of our manuscript, we have taken your recommendation on board. In order to address this, we have summerised the manuscript while including the additional detail needed to address both the editor and reviewer 2. We feel by only addressing the new material, the inherent issues faced during allergy diagnosis and treatment was not fully explained. During the initial review process, major changes were suggested to better clarify not only the problems faced but the new data available, which highlight more clearly the benefits of molecular diagnostics in combination with recombinant and hypoallergenic variants. These have now been fully addressed in the revised manuscript.   

Reviewer 2 Report

Line 41. The CDC report cited states that nine percent of U.S. children under the age 18 suffered from hay fever in the past 12 months, 11% from respiratory allergies, 6% from food allergies, and 12% from skin allergies. It does not give global estimates or the 250 million figure used in this manuscript. Asthma affects 300M people worldwide and recent estimates allergic rhinitis show it would be twice that number.

Line 52. Reference 9 contains the statement " Food hypersensitivities develop in genetically pre- disposed individuals" giving no reference or discussion of food allergy and family history.  There are studies of genetics and food allergy.  While they show an increased risk for subjects with a family history of food allergy about 70% of food allergic subject have no family history (similar other types of allergic disease).

Reference 11 does not present any evidence to show that leaky gut syndrome allows the passage of allergens. There is an uncorroborated paper on an association of leaky gut with asthma which if true would probably be more associated with the loss of barrier function elsewhere. Leaky gut syndrome has been studied with respect to autoimmune disease and stimulation with gut microbial antigens with no suggestion of food allergy.

Line 72 and c. Reference 14 reports a 6% prevalence of self-reported egg allergy and then goes on to qualify this proposing that standardised methods are required for true measurements. Reference 15 studies the development of egg tolerance not prevalence and cites 1-2% as the commonly found prevalence. Where is the 9% in these references?

Line 76. The health Nuts study has been misrepresented. It should be made clear that 9% refers to allergies found at 12 months (15) and not a persistent allergy with the prevalence being 1.2% at 4 years (ref 18).

Line 108-110. It not understood what this non-sentence is meant to mean. Reference 25 shows that cooked eggs have less allergenicity that is probably related to structure but this was not demonstrated and was written in 2008.  Reference 28 was published in 2020 and was consensus document that included WHO but says little about structure.

The component resolved diagnosis is poorly presented and not complete. A comparison of the different studies is required including those not cited here.

Table 2 should show the IUIS allergen denominations (especially since the denominations are used in the text). Molecular weights do not have units.

Need to explain what PMAD@ stands for

Figure 3 is completely unnecessary

Figure 4 is both unnecessary and incorrect. Cell mediate immunity refers to T-cell mediated phenomena.

Figure 5 does give a citation for the epitopes or discuss their veracity. Reference 103 is about making recombinant Gal d 1, 2 and 3 egg allergens. It seems that epitopes shown were constructed from information of Jimenez- Saiz et al, doi: 10.1002/mnfr.201300442 which examined a single pool of allergic serum. Apart from possibly representing only one person's binding pattern the individual "epitopes" shown were not tested for their ability block IgE binding to lysozyme (although a large digest product that contained all the residues linked by disulphide bonds was). The percentage of IgE compared to the whole allergen is thus unknown.

Reference 103 and others not included in this review would be relevant to a properly constructed overview of the use of PMAD@ for the diagnosis and prognosis of egg allergy and would be relevant to the construction of egg hypoallergens.

Reference 94. There is only a title given in the list (THE CURRENT STATE OF ORAL IMMUNOTHERAPY (OIT) FOR THE TREATMENT OF FOOD ALLERGY). It wasn't found in a search of the literature.

Contrary to what is presented here the use of OIT for egg allergy is considered to be a very promising treatment with widespread use. There are more than several very good publications on the specific studies for egg allergy and comparisons (favourable) with OIT for other food allergens.

Author Response

Thank you for taking the time to review our manuscript and once again providing feedback. In response to your comments minor changes were made to the entire manuscript. In addition, the following comment was addressed. Please find below brief points to all of the general comments (reviewers’ comments in blue, our response in black). These have been highlighted in yellow within the manuscript

Line 41. The CDC report cited states that nine percent of U.S. children under the age 18 suffered from hay fever in the past 12 months, 11% from respiratory allergies, 6% from food allergies, and 12% from skin allergies. It does not give global estimates or the 250 million figure used in this manuscript. Asthma affects 300M people worldwide and recent estimates allergic rhinitis show it would be twice that number.

            The global estimate of 250 million individuals being affected by food allergies is from WAO 2014. We sincerely apologise for this error due to the extensive major changes made during the prior revisions, it seems EndNote did not update/configure all the intext citations to match the reference list. This has now been corrected and the references re-checked.

Pawankar, R. Allergic diseases and asthma: a global public health concern and a call to action. World Allergy Organ J 7, 1–3 (2014). https://doi.org/10.1186/1939-4551-7-12

            Pawankar states “Globally, 300 million people suffer from asthma and about 200 to 250 million people suffer from food allergies”

Line 52. Reference 9 contains the statement " Food hypersensitivities develop in genetically pre- disposed individuals" giving no reference or discussion of food allergy and family history.  There are studies of genetics and food allergy.  While they show an increased risk for subjects with a family history of food allergy about 70% of food allergic subject have no family history (similar other types of allergic disease).

     This paragraph now reads “Genetics, however, may be the strongest link in further understanding this phenomenon [7]. It was found that a child has up to 75% chance of developing an allergy given that both parents also suffer from allergies [8].Genetically predisposed individuals have an inherited tendency to develop a food allergy with The HealthNut study re-defining “high risk” patients as having two or more allergic family members [9]. However, it is possible to develop allergies even if there is no family history, with research indicating only a slight increase in food allergy of infants with one allergic member in comparison to those with no family history [9].This suggests other factors may also be involved resulting in various theories including the hygiene hypothesis [10]. This hypothesis proposes that the protective effect of microbial contact in the early stages of life is lost due to increased hygiene and extensive use of antibiotics within Western culture. Thereupon, an allergy develops as the immune system lacks predisposition to infection where the microbial stimulus necessary for normal maturity of the immune system is lost [11]. Population based study of infants suggest unlike peanut allergies, egg allergies are heavily motivated by environmental factors [9]. Epidemiological research indicate the hygiene hypothesis to be a complex interaction of many factors and such further large scale population based studies are needed to investigate genetic susceptibility, timing, allergy phenotypes and various environmental exposures [12].

We have now added the following reference:

Koplin, J.J.; Allen, K.J.; Gurrin, L.C.; Peters, R.L.; Lowe, A.J.; Tang, M.L.K.; Dharmage, S.C.; HealthNuts Study, T. The impact of family history of allergy on risk of food allergy: a population-based study of infants. Int J Environ Res Public Health 2013, 10, 5364-5377, doi:10.3390/ijerph10115364.

von Mutius, E. Allergies, infections and the hygiene hypothesis – The epidemiological evidence. Immunobiology 2007, 212, 433-439, doi:https://doi.org/10.1016/j.imbio.2007.03.002.

Reference 11 does not present any evidence to show that leaky gut syndrome allows the passage of allergens. There is an uncorroborated paper on an association of leaky gut with asthma which if true would probably be more associated with the loss of barrier function elsewhere. Leaky gut syndrome has been studied with respect to autoimmune disease and stimulation with gut microbial antigens with no suggestion of food allergy.

This section was removed in order to condense the manuscript as per feedback from Reviewer 1. Liu et al discussed the possibility of food antigens crossing the lamina propria of the gastrointestinal tract. This occurs as antigens cross the permeable lamina propria of the gastrointestinal tract, if a food antigen enters the lymphocyte harboring lamina propria the APCs may present them to T-cells thus producing sIgE. However, given that this required further detail and explanation it was removed in order to better summarize the manuscript.

Liu, Z., Li, N. and Neu, J. (2005), Tight junctions, leaky intestines, and paediatric diseases. Acta Pædiatrica, 94: 386-393. doi:10.1111/j.1651-2227.2005.tb01904.x

Line 72 and c. Reference 14 reports a 6% prevalence of self-reported egg allergy and then goes on to qualify this proposing that standardised methods are required for true measurements. Reference 15 studies the development of egg tolerance not prevalence and cites 1-2% as the commonly found prevalence. Where is the 9% in these references?

This sentence now reads “The hypersensitivity to Gallus gallus (chicken) egg is a pervasive condition ordinarily affecting up to 9% of the children worldwide [16,17]”. The in-text citation [17] refers to the following publication:

Wei-Liang Tan, J.; Valerio, C.; Barnes, E.H.; Turner, P.J.; Van Asperen, P.A.; Kakakios, A.M.; Campbell, D.E. A randomized trial of egg introduction from 4 months of age in infants at risk for egg allergy. Journal of Allergy and Clinical Immunology 2017, 139, 1621-1628.e1628, doi:https://doi.org/10.1016/j.jaci.2016.08.035.

Wei-Liang et al states that “Egg allergy is common, with an estimated prevalence of 1% to 9% around the world”.

Line 76. The health Nuts study has been misrepresented. It should be made clear that 9% refers to allergies found at 12 months (15) and not a persistent allergy with the prevalence being 1.2% at 4 years (ref 18).

             This sentence was re-written to clarify the prevalence of egg allergy in 12-month-old children to be 9.5%.

Line 108-110. It not understood what this non-sentence is meant to mean. Reference 25 shows that cooked eggs have less allergenicity that is probably related to structure but this was not demonstrated and was written in 2008.  Reference 28 was published in 2020 and was consensus document that included WHO but says little about structure.

                  This section was removed in order to condense the manuscript as per feedback from Reviewer 1. This information was summarized into point 6 ‘Production and Need for Recombinant and Hypoallergenic Egg Allergens’ to clarify the importance of IgE binding epitopes. Information from just WAO included to reiterate the link between IgE binding epitopes and egg allergy phenotypes, therefore proving target during site-directed mutagenesis. The original source discussing structure is now included and listed below.

Chatchatee, P.; Jarvinen, K.M.; Bardina, L.; Beyer, K.; Sampson, H.A. Identification of IgE- and IgG-binding epitopes on alpha(s1)-casein: differences in patients with persistent and transient cow's milk allergy. J Allergy Clin Immunol 2001, 107, 379-383, doi:10.1067/mai.2001.112372.

Need to explain what PMAD@ stands for

Page 10 line 403 reads “The growing availability of allergenic molecules has introduced a new phase in diagnostics now termed precision allergy molecular diagnostic applications (PAMD@) allowing for greater management of the allergic disease”.

Figure 4 is both unnecessary and incorrect. Cell mediate immunity refers to T-cell mediated phenomena.

Figure 4 changed to ‘the allergenic response’ as per the editor’s feedback.

Table 2 should show the IUIS allergen denominations (especially since the denominations are used in the text). Molecular weights do not have units.

The allergens are now listed under IUIS allergen denominations. The column for molecular weight now indicates SDS-PAGE and therefore indicated by kDa.

Reference 94. There is only a title given in the list (THE CURRENT STATE OF ORAL IMMUNOTHERAPY (OIT) FOR THE TREATMENT OF FOOD ALLERGY). It wasn't found in a search of the literature.

Again, we sincerely apologise for this, it was an error when configuring the final document with EndNote. Though the URL was attached to this reference it was not shown in the final bibliography. We have manually edited this reference in EndNote to show the URL. The following is the correct reference:

AAAAI. The current state of oral immunotherapy (OIT) for the treatment of food allergy. Availabe online: https://www.aaaai.org/conditions-and-treatments/library/allergy-library/oit

Figure 5 does give a citation for the epitopes or discuss their veracity. Reference 103 is about making recombinant Gal d 1, 2 and 3 egg allergens. It seems that epitopes shown were constructed from information of Jimenez- Saiz et al, doi: 10.1002/mnfr.201300442 which examined a single pool of allergic serum. Apart from possibly representing only one person's binding pattern the individual "epitopes" shown were not tested for their ability block IgE binding to lysozyme (although a large digest product that contained all the residues linked by disulphide bonds was). The percentage of IgE compared to the whole allergen is thus unknown.

Figure 5 is now referenced. The data regarding the placement of the IgE binding epitopes and disulphide bridges are from the following references listed below. Gal d 4 was only used as an example to highlight the disulphide bridges within the IgE binding epitopes that provide an ideal target for site-directed mutagenesis, allowing for the production of hypoallergens.

UniProt. UniProtKB - P00698 (LYSC_CHICK). Availabe online: https://www.uniprot.org/uniprot/P00698

Matsuo, H.; Yokooji, T.; Taogoshi, T. Common food allergens and their IgE-binding epitopes. Allergology international : official journal of the Japanese Society of Allergology 2015, 64, 332-343, doi:10.1016/j.alit.2015.06.009.

Haeffner-Gormley, L.; Parente, L.; Wetlaufer, D.B. Use of proline-specific endopeptidase in the isolation of all four "native" disulfides of hen egg white lysozyme. Int J Pept Protein Res 1985, 26, 83-91, doi:10.1111/j.1399-3011.1985.tb03181.x.

Figure 3 is completely unnecessary

We feel that both figure 3 and 4 are both important due to the lack of such high-quality figures currently available from reputable sources. In addition, neither the Editor or eviewer 1 have requested its removal.

The component resolved diagnosis is poorly presented and not complete. A comparison of the different studies is required including those not cited here.

Unfortunately, in order to condense the manuscript, as suggested by Reviewer 1, we were unable to further discuss and compare CRD diagnostics in food allergy. However, the following reference is now included to direct the reader to a detailed systematic review conducted by Kim and colleagues. Within the revised manuscript, the need for further diagnostic accuracy tests are highlighted.

Flores Kim, J.; McCleary, N.; Nwaru, B.I.; Stoddart, A.; Sheikh, A. Diagnostic accuracy, risk assessment, and cost-effectiveness of component-resolved diagnostics for food allergy: A systematic review. Allergy 2018, 73, 1609-1621, doi:10.1111/all.13399.

Reference 103 and others not included in this review would be relevant to a properly constructed overview of the use of PMAD@ for the diagnosis and prognosis of egg allergy and would be relevant to the construction of egg hypoallergens.

The manuscript now further discusses the work conducted by us in the following publications:

Dhanapala, P.; Withanage-Dona, D.; Tang, M.L.; Doran, T.; Suphioglu, C. Hypoallergenic Variant of the Major Egg White Allergen Gal d 1 Produced by Disruption of Cysteine Bridges. Nutrients 2017, 9, doi:10.3390/nu9020171.

Lemon-Mule, H.; Sampson, H.A.; Sicherer, S.H.; Shreffler, W.G.; Noone, S.; Nowak-Wegrzyn, A. Immunologic changes in children with egg allergy ingesting extensively heated egg. J Allergy Clin Immunol 2008, 122, 977-983 e971, doi:10.1016/j.jaci.2008.09.007.

Contrary to what is presented here the use of OIT for egg allergy is considered to be a very promising treatment with widespread use. There are more than several very good publications on the specific studies for egg allergy and comparisons (favourable) with OIT for other food allergens.

The following publications were used to further highlight the benefits of OIT:

Tang, M.L.K.; Mullins, R.J. Food allergy: is prevalence increasing? Internal Medicine Journal 2017, 47, 256-261, doi:10.1111/imj.13362

Tang, M.L.K.; Martino, D.J. Oral immunotherapy and tolerance induction in childhood. Pediatric Allergy and Immunology 2013, 24, 512-520, doi:10.1111/pai.12100.

Tang, M.L.; Ponsonby, A.L.; Orsini, F.; Tey, D.; Robinson, M.; Su, E.L.; Licciardi, P.; Burks, W.; Donath, S. Administration of a probiotic with peanut oral immunotherapy: A randomized trial. J Allergy Clin Immunol 2015, 135, 737-744.e738, doi:10.1016/j.jaci.2014.11.034.

Ismail, I.H.; Tang, M.L. Oral immunotherapy for the treatment of food allergy. Isr Med Assoc J 2012, 14, 63-69.

Buchanan, A.D.; Green, T.D.; Jones, S.M.; Scurlock, A.M.; Christie, L.; Althage, K.A.; Steele, P.H.; Pons, L.; Helm, R.M.; Lee, L.A., et al. Egg oral immunotherapy in nonanaphylactic children with egg allergy. J Allergy Clin Immunol 2007, 119, 199-205, doi:10.1016/j.jaci.2006.09.016.

Vickery, B.P.; Pons, L.; Kulis, M.; Steele, P.; Jones, S.M.; Burks, A.W. Individualized IgE-based dosing of egg oral immunotherapy and the development of tolerance. Ann Allergy Asthma Immunol 2010, 105, 444-450, doi:10.1016/j.anai.2010.09.030.